# Altered transcriptomic immune responses of maintenance hemodialysis patients to the COVID-19 mRNA vaccine

Yi-Shin Chang[1,2†], Kai Huang[1,2†], Jessica M Lee[1,3], Christen L Vagts[1], Christian Ascoli[1], Md-Ruhul Amin[1], Mahmood Ghassemi[1], Claudia M Lora[1], Russell Edafetanure-Ibeh[1], Yue Huang[1], Ruth A Cherian[1], Nandini Sarup[1], Samantha R Warpecha[1], Sunghyun Hwang[1], Rhea Goel[1], Benjamin A Turturice[1,3,4], Cody Schott[1,3,5], Montserrat Hernandez[1], Yang Chen[1,6], Julianne Jorgensen[1,2], Wangfei Wang[1,2], Mladen Rasic[1,2], Richard M Novak[1], Patricia W Finn[1,2,3*‡], David L Perkins[1,2,6‡]

[1]Department of Medicine, University of Illinois at Chicago, Chicago, United States; [2]Department of Bioengineering, University of Illinois at Chicago, Chicago, United States; [3]Department of Microbiology and Immunology, University of Illinois at Chicago, Chicago, United States; [4]Department of Medicine, Stanford University, Palo Alto, United States; [5]Department of Medicine, University of Colorado Denver, Aurora, United States; [6]Department of Biological Sciences, University of Illinois at Chicago, Chicago, United States

*For correspondence:
pwfinn@salud.unm.edu

†These authors contributed equally to this work
‡These authors also contributed equally to this work

## Abstract

**Background:** End-stage renal disease (ESRD) patients experience immune compromise characterized by complex alterations of both innate and adaptive immunity, and results in higher susceptibility to infection and lower response to vaccination. This immune compromise, coupled with greater risk of exposure to infectious disease at hemodialysis (HD) centers, underscores the need for examination of the immune response to the COVID-19 mRNA-based vaccines.

**Methods:** The immune response to the COVID-19 BNT162b2 mRNA vaccine was assessed in 20 HD patients and cohort-matched controls. RNA sequencing of peripheral blood mononuclear cells was performed longitudinally before and after each vaccination dose for a total of six time points per subject. Anti-spike antibody levels were quantified prior to the first vaccination dose (V1D0) and 7 d after the second dose (V2D7) using anti-spike IgG titers and antibody neutralization assays. Anti-spike IgG titers were additionally quantified 6 mo after initial vaccination. Clinical history and lab values in HD patients were obtained to identify predictors of vaccination response.

**Results:** Transcriptomic analyses demonstrated differing time courses of immune responses, with prolonged myeloid cell activity in HD at 1 wk after the first vaccination dose. HD also demonstrated decreased metabolic activity and decreased antigen presentation compared to controls after the second vaccination dose. Anti-spike IgG titers and neutralizing function were substantially elevated in both controls and HD at V2D7, with a small but significant reduction in titers in HD groups (p<0.05). Anti-spike IgG remained elevated above baseline at 6 mo in both subject groups. Anti-spike IgG titers at V2D7 were highly predictive of 6-month titer levels. Transcriptomic biomarkers after the second vaccination dose and clinical biomarkers including ferritin levels were found to be predictive of antibody development.

**Conclusions:** Overall, we demonstrate differing time courses of immune responses to the BTN162b2 mRNA COVID-19 vaccination in maintenance HD subjects comparable to healthy controls and

identify transcriptomic and clinical predictors of anti-spike IgG titers in HD. Analyzing vaccination as an in vivo perturbation, our results warrant further characterization of the immune dysregulation of ESRD.

**Funding:** F30HD102093, F30HL151182, T32HL144909, R01HL138628. This research has been funded by the University of Illinois at Chicago Center for Clinical and Translational Science (CCTS) award UL1TR002003.

## Editor's evaluation

Chang et al. demonstrate through their findings that COVID-19 mRNA vaccination of hemodialysis patients produces no significant difference in antibody levels achieved across the vaccination series. However, they find that T-cell responses may be delayed in hemodialysis patients as they have lower activation of T-cell genes than healthy controls. The RNA sequencing evidence is solid. However, they lack data on a clinical correlation to T-cell responses.

## Introduction

End-stage renal disease (ESRD) is the most advanced stage of chronic kidney disease (CKD), with prevalence in the United States reaching 809,000 in 2019 (*Johansen et al., 2022*). The most used form of renal replacement therapy for ESRD patients in the United States is hemodialysis (HD). Despite significant improvements in HD technology, the mortality rate in ESRD patients is still as high as 20% annually (*Williams et al., 2004*), with infections being the most common cause of hospitalization and mortality after cardiovascular disease (*Kato et al., 2008*). The immunocompromised state of ESRD is characterized by simultaneous immunodepression due to the impact of uremic milieu on immunocompetent cells and immunoactivation due to the accumulation of proinflammatory cytokines (*Kato et al., 2008*). There are alterations to both innate and adaptive immunity, including elevated levels of mannose-binding lectin (*Satomura et al., 2002*), impaired maturation of monocytes and dendritic cells (*Satomura et al., 2002*; *Lim et al., 2007*), increased B cell apoptosis (*Fernández-Fresnedo et al., 2000*), and decreased T-cell proliferation with elevated Th1/Th2 ratio (*Stenvinkel et al., 2005*).

Studies of genome-wide expression (i.e., transcriptome) profiles of peripheral blood mononuclear cells (PBMCs) in ESRD demonstrate a complex picture of immune alterations. One study found upregulation of genes involved in the complement and oxidative metabolism pathways, and downregulation of genes associated with the clathrin-coated vesicle endosomal pathway and T-cell receptor signaling (*Scherer et al., 2013*). Two other studies have demonstrated impaired expression of genes involved in oxidative phosphorylation and mitochondrial function (*Granata et al., 2009*; *Liu et al., 2002*). A study identifying a group of inflammatory genes playing a causative role in oxidative stress in dialysis patients showed unique gene expression alterations in maintenance HD patients compared to un-dialyzed CKD patients and compared to patients undergoing peritoneal dialysis (*Zaza et al., 2008*). These studies indicate a range of immune pathways that may impair vaccination response, and further suggest that dialysis leads to unique immune profile alterations. These alterations may lead to higher susceptibility to infection and lower response to vaccination (*Ghadiani et al., 2012*). For example, while more than 90% of patients without CKD develop protective antibodies against HBV after vaccination, only 50–60% of patients with ESRD seroconvert. There have also been higher vaccination failure rates demonstrated against influenza virus, *Clostridium tetani*, and *Corynebacterium diphtheriae* in ESRD (*Eleftheriadis et al., 2007*).

Understanding the immune response to vaccines in ESRD is particularly important for the new mRNA vaccines developed in response to the COVID-19 pandemic. The COVID-19 mRNA-based vaccines, BNT162b2 and mRNA-1273, have proven to be efficacious in non-immunocompromised individuals, with initial reports showing 95 and 94.1% reduction of COVID-19 disease in recipients (*Baden et al., 2021*; *Polack et al., 2020*). However, certain immunosuppressed populations remain at risk of infection. Given the widespread transmission of COVID-19, detailed assessments of degree, duration, and determinants of immune protection conferred by these vaccines are vitally needed in immunocompromised patient populations including those with ESRD.

While recent studies of the SARS-CoV-2 BTN162b2 vaccine in HD demonstrate high levels of seroconversion ranging from 84 to 96% (*Grupper et al., 2021*; *Jahn et al., 2021*; *Attias et al., 2021*;

*Anand et al., 2021*), they also demonstrate quantitatively reduced SARS-CoV-2 IgG antibodies. We posit that characterization of the transcriptomic underpinnings of antibody titer development on a continuous scale may identify biomarkers for weaker or less durable immune protection in this population. Furthermore, transcriptomic analyses may identify strategies for the development of new, effective vaccines against other infectious diseases for this population. Thus, we characterized the immune response of the HD population to the COVID-19 mRNA-based BNT162b2 vaccine using mRNA sequencing, anti-spike antibody ELISA, and neutralization titers across multiple time points. We then integrated these data to identify transcriptomic and clinical determinants of the humoral immune response in HD patients.

## Methods

### Study population and sample acquisition

The study was approved by the University of Illinois at Chicago IRB (#2018-1038) Ethics Review Committee. Maintenance HD patients undergoing vaccination with the BNT162b2 mRNA COVID-19 vaccine in February 2021 were recruited from the outpatient HD unit at the University of Illinois Hospital (UIH) in Chicago, IL. Control subjects consisted of UIH employees undergoing BNT162b2 mRNA COVID-19 vaccination at UIH from December 2020 to January 2021 with no self-reported history of kidney disease or immune disorders. All subjects provided informed consent for participation in this research and for publication of results. A subset of control subjects matched for age, gender, and COVID-19 history was also analyzed for this study. Blood was collected at 0–48 hr prior to and at multiple time points after both the first (V1) and second vaccination doses (V2), which were administered 3 wk apart. Control samples were collected prior to each vaccination dose (D0) and at 1 d (D1) and 7 d (D7) after each dose, corresponding to six time points: V1D0, V1D1, V1D7, V2D0, V2D1, and V2D7. Blood was collected from HD subjects prior to each vaccination dose and at 2 d (D2) and 7 d after each dose, corresponding to six time points: V1D0, V1D2, V1D7, V2D0, V2D2, and V2D7. A final blood sample was drawn 6 mo after initial vaccination (M6) for measurement of antibody titers, prior to additional vaccination doses. Serum and PBMCs were extracted within 2 hr of blood collection, then stored at –80°C. PBMCs were extracted using density gradient centrifugation at 400 × *g* with Ficoll–Paque PLUS. The extracted buffy coat was stored in RNAlater (Invitrogen).

### Clinical and demographic characterization

Demographic and clinical data was collected from the electronic health record (EHR) for HD subjects, including medical diagnoses, medications, and laboratory values. Laboratory values included monthly SARS-CoV-2 test results, as well as urea reduction ratio (URR, a measure of dialysis adequacy), hemoglobin (Hgb), ferritin, transferrin saturation, albumin levels, white blood cell (WBC) count, and WBC differential counts obtained during standard of care monthly blood draws for the 3 mo preceding vaccination. Within our analyses, ferritin was coded as either low risk (200–1200 ng/ml) or high risk (<200 or >1200 ng/ml) since ferritin levels 200–1200 ng/ml have been shown to be associated with lowest all-cause mortality in HD patients (*Kalantar-Zadeh et al., 2005*). Baseline clinical lab values were calculated as the median of three lab values across the 3 mo prior to vaccination. Demographic and clinical data was collected from a medical questionnaire at time of consent for control subjects, and included medical history, medications, and self-reported prior SARS-CoV-2-positive test results.

### RNA extraction and RNA sequencing (RNAseq)

RNA sequencing was performed on PBMCs at all V1 and V2 time points for all subjects for whom RNA libraries were successfully built at ≥5 time points. PBMCs stored in RNAlater were thawed and diluted 1:1 with 1× phosphate buffered saline. The mixture was then pelleted and RNA was extracted using the PureLink RNA Mini kit (Invitrogen). DNase treatment to remove genomic DNA contamination was performed using either the PureLink DNase kit or DNA-free DNA Removal Kit (Invitrogen). Purified RNA in sterile water was stored at –80°C. Each RNA sample was quantified using the Qubit RNA High Sensitivity kit (Invitrogen) and Bioanalyzer RNA Pico kit (Agilent) with RIN ≥ 8.

For library construction, 50 ng of RNA from each sample was aliquoted in 96-well plates. Libraries were generated using the NEBNext Ultra II Directional RNA Library Prep Kit for Illumina with the optional NEBNext Poly(A) mRNA Magnetic Isolation Module (New England BioLabs). Each individual

sample library was barcoded during PCR amplification using unique dual indexed i5 and i7 primers from the NEBNext Multiplex Oligos for Illumina kit. Each sample library was quantified using the Qubit DNA High Sensitivity kit and Bioanalyzer DNA High Sensitivity kit. Samples were then pooled and sequenced using the MiSeq Nano V2 kit (Illumina) to check read proportions between samples. Samples with lower-than-expected percentage of reads detected were supplemented with an additional spike-in of sample library to the main pool. The supplemented pooled library was sequenced again using the MiSeq Nano V2 kit to verify adequate adjustment. The finalized library was sequenced using a NovaSeq S2 flow cell configured for 75 bp paired end output.

## Differential gene expression analysis

Raw demultiplexed reads were filtered using fastp to remove adapters and short reads (*Chen et al., 2018*). Trimmed reads were then quantified using the Salmon pipeline with an hg38 reference transcriptome index (*Patro et al., 2017*). Quantified data was imported into R using the tximeta package (*Love et al., 2020*) to convert Salmon quantification and index data to a count matrix. Transcript names were extracted and matched using Entrez IDs with the AnnotationHub package (*Love et al., 2020*). This finalized count matrix was then imported into a DESeqDataset object and normalized using the variance stabilizing transformation in DESeq2.

The *DESeq2* R package was used to identify genes that were differentially expressed at each time point after vaccination for each subject group. Specifically, we implemented a design incorporating group-specific condition effects with individual subjects nested within groups. We performed the classical *Deseq2* workflow of estimation of size factors, estimation of dispersion, and negative binomial GLM fitting for $\beta_i$ and Wald statistics, increasing the maximum number of iterations for estimation of the negative binomial distribution to 500. We then generated contrasts to obtain differentially expressed genes for controls at V1D1 and V1D7 (compared to V1D0), and at V2D1 and V2D7 (compared to V2D0). Differentially expressed genes for HD were similarly obtained at V1D2 and V1D7 (compared to V1D0), and at V2D2 and V2D7 (compared to V2D0). We also directly compared gene expression between controls and HD at V1D7 and at V2D7. The significance threshold to determine differential expression was FDR-adjusted ($p < 0.05$).

## Anti-spike (trimer) IgG titer quantification

The Human SARS-CoV-2 Spike (Trimer) IgG ELISA Kit from Invitrogen was used to quantitate IgG to the SARS-CoV-2 spike protein in serum samples at V1D0, V2D7, and M6 time points. All samples were initially diluted 1:100 (in addition to the 1:10 assay buffer dilution on the 96-well plate) and assayed in duplicate, with twofold serial dilution of the 150,000 units/ml standard control in duplicate for relative quantification. Absorbance at 450 nm was quantified using a *Spark* multimode microplate reader. Samples that produced signals greater than the upper limit of the standard curve were diluted 1:2000 and assayed again. IgG concentration was calculated by fitting four-parameter logistic curves to the standard controls and taking the average concentrations of duplicates.

## Antibody neutralization assays

Neutralization assays were performed on serum samples from V1D0 and V2D7 using SARS-CoV-2 pseudotyped virus (pseudovirus). To produce pseudoviruses, an expression plasmid bearing codon-optimized SARS-CoV-2 full-length S plasmid was co-transfected into HEK293T cells using the SARS-CoV-2 spike-pseudotyped lentiviral particle Kit (BEI # R-52948). The cell supernatants were collected 72 hr after transfection, divided into aliquots and cryopreserved at −80°C.

To titrate the pseudovirus, $5 \times 10^3$ 293T-ACE2 cells were seeded per well in a 96-well plate in DMEM containing 10% FBS and 1% penicillin streptomycin. Twenty-four hours later, the pseudovirus was diluted 1:10, followed by fivefold serial dilutions for a total of nine dilutions, with each dilution performed in six replicate wells. After incubation at 37°C and 5% (vol/vol) $CO_2$ for 72 hr, the luciferase substrate was added to the 96-well plate for chemiluminescence detection. The 50% tissue culture infectious dose (TCID50) of the pseudovirus was calculated according to the Reed–Muench method in the titration macro template (*Matumoto, 1949*).

Neutralization activity against SARS-2-CoV was measured in a single-round-of-infection assay with pseudoviruses as previously described (*Nie et al., 2020*). $5 \times 10^3$ 293T-ACE2 cells were seeded per well in a 96-well plate. Twenty-four hours later, serial dilutions of the serum samples were performed,

incubated for 1 hr at 37°C with ~1000 TCID50/ml of pseudovirus, then added to monolayers of ACE2-overexpressing 293T cells in quadruplicate. The cell control with cells alone and the virus control (VC) with pseudovirus were set up in each plate. The target cells were incubated for 65–72 hr at 37°C and 5% (vol/vol) $CO_2$. Then, 50 µl of Bright-Glo, reconstituted following the manufacturer's instructions, was added to each well of the 96-well plate and incubated for 5 min at room temperature. The 96-well plate was read by a 96-well luminescence plate reader (Tecan Genius Pro plate reader) (*Ferrara and Temperton, 2018*). Percent neutralization was calculated as 100 * ([Virus-only control] – [Virus plus serum])/[Virus-only control], and neutralizing titer levels are reported as the serum dilution required to achieve 50% neutralization (50% inhibitory dilution [ID50]) (*Pegu et al., 2021*). The input dilution of serum was 1:20, thus 20 is the lower limit of quantification.

## BTM module enrichment analysis

Gene set enrichment analysis was performed for each contrast generated in the DESeq2 analysis above using blood transcription module (BTMs) gene sets (*Li et al., 2014*). BTMs with FDR-adjusted $p<0.05$ were considered significantly enriched. Enriched BTMs were further characterized using the distribution of Wald statistics of membership genes from DESeq2. To summarize BTM analyses, BTMs were categorized into different families: B cells, cell cycle, dendritic cell/antigen presentation, type I interferon (IFN type I), myeloid activity/inflammation/ T/NK cells, and 'others' (*Braun et al., 2018*). The percentage of BTMs in each BTM family with significant enrichment at each time point was then quantified over time.

## Statistical analysis of antibody response

To determine the effect of vaccination on anti-spike IgG titers at V2D7 and M6, Kruskal–Wallis tests were performed separately for HD subjects and controls. For each group, anti-spike IgG titer levels were compared to assess for the significant effect of time (V1D0, V2D7, M6), and Wilcoxon rank-sum tests were performed with FDR correction to assess significant differences between each pair of time points (V2D7 vs. V1D0, M6 vs. V1D0, M6 vs. V2D7). To determine the effect of vaccination on anti-body neutralization activity (ID50) at V2D7, Wilcoxon rank-sum tests were performed for each group to compare V2D7 vs. V1D0.

Linear models were constructed to establish the effect of prior SARS-CoV-2 infection and subject group on anti-spike IgG titer development at V2D7 and M6 and neutralization activity at V2D7. Specifically, log-transformed V2D7 anti-spike IgG titers or V2D7 neutralization activity (ID50) were modeled as the dependent variable, with subject group (HD or controls), log-transformed V1D0 anti-spike IgG titers or V1D0 neutralization activity (ID50), gender, age, race, and ethnicity as independent predictors. To determine predictors of anti-spike IgG at 6 mo, a linear model was constructed with the log-transformed M6 anti-spike IgG titers as the dependent variable, and V2D7 anti-spike IgG titers, SARS-CoV-2 history, gender, age, race, and ethnicity as independent predictors.

## Identification of BTM and clinical predictors of Ab response in HD

BTM predictors of antibody response in HD were identified by first calculating a representative expression level of each BTM per sample, which we will refer to as the eigengene. Specifically, the first principal component of each BTM was calculated using DESeq2-derived variance-stabilized gene counts from each module's member genes across the HD V1 time points, and then across the HD V2 time points. Signs (positive or negative) were assigned to the eigengenes such that samples with higher expression of member genes in a BTM would be given a positive sign, while those with lower overall gene expression would be given a negative sign. This was accomplished for each BTM by (1) computing the median gene expression level across membership genes in a given BTM for each sample, (2) computing the Pearson correlation between the eigengene of the BTM and the median gene expression level across all samples, and (3) multiplying the eigengene of the BTM by –1 if the correlation was negative.

Subsequently, we constructed linear models with log-transformed anti-spike IgG at V2D7 as the dependent variable and change in BTM eigengene expression after vaccination as the independent variable, controlling for SARS-CoV-2 history. Separate models were constructed for each BTM that was enriched at each time point after vaccination in HD (V1D2 vs. V1D0, V1D7 vs. V1D0, V2D2 vs. V2D0, V2D0 V2D7). Change in BTM expression was calculated as the BTM eigengene after vaccination minus

**Table 1.** Demographic and clinical data for maintenance hemodialysis and control subjects.

| | Hemodialysis | Control | p-value |
|---|---|---|---|
| Total # of subjects | 20 | 20 | |
| **Gender** | | | |
| Male | 11 | 10 | 1.0 |
| Female | 9 | 10 | 1.0 |
| **Age (mean (sd))** | 54 (12) | 54 (13) | 0.98 |
| **Race/ethnicity** | | | |
| Black/African American | 10 | 3 | 0.041 |
| Asian/Pacific Islander | 1 | 2 | 1.0 |
| White/Caucasian | 2 | 8 | 0.067 |
| Hispanic/Latinx | 7 | 6 | 1.0 |
| Other | 0 | 1 | 1.0 |
| **BMI, kg/m2 (mean (sd))** | 27.8 (5.1) | 28.7 (6.4) | 0.61 |
| **Medical history** | | | |
| Diabetes | 11 | 1 | 0.0012 |
| Hypertension | 18 | 4 | <0.001 |
| Other CV disease[*] | 9 | 0 | 0.0012 |
| Dyslipidemia | 10 | 0 | <0.001 |
| Autoimmune disease[†] | 3 | 0 | 0.23 |
| Immunosuppression[‡] | 1 | 0 | 1.0 |
| Active malignancy[§] | 1 | 0 | 1.0 |
| **Positive COVID-19 history** | 8 | 5 | 0.5 |

[*]Includes coronary artery disease (CAD), congestive heart failure (CHF), atrial fibrillation (AF), peripheral vascular disease (PVD), and cerebral vascular accent (CVA).

[†]Includes systemic lupus erythematosus (SLE), immune thrombocytopenic purpura (ITP), and microscopic polyangiitis (MPA).

[‡]Hydroxychloroquine.

[§]Defined as malignancy requiring treatment in the last 6 mo; one patient with papillary thyroid cancer requiring thyroidectomy, no systemic treatment required.

the BTM eigengene before vaccination. p-values were FDR-adjusted across number of enriched BTMs per time point.

Additionally, baseline clinical laboratory values predictive of antibody response in the HD subjects were identified. Linear models were separately constructed using URR, ferritin (high risk vs. low risk), transferrin saturation, hemoglobin, and WBC count to predict log-transformed anti-spike IgG at V2D7 and M6.

Finally, clinical laboratory values responding to vaccination that predicted antibody titer response in the HD subjects were identified. Linear models were separately constructed using log-fold change (LFC) from baseline measurements of ferritin (continuous instead of binarized low- and high-risk), transferrin saturation, and WBC count to predict log-transformed anti-spike IgG titers at V2D7 and M6.

**Table 2.** Baseline clinical lab values for maintenance hemodialysis (HD) patients.

| | Normal range | Mean (SD) |
|---|---|---|
| **Kidney/HD status** | | |
| Urea reduction ratio | - | 0.74 (0.052) |
| Months on HD | - | 46 (44) |
| **Iron** | | |
| Ferritin (ng/ml) | 10–259 | 838 (550) * |
| % Transferrin saturation | 25–50 | 38 (13) |
| Albumin | 3.4–5 | 4.1 (0.40) |
| **CBC** | | |
| White blood cells (k/ul) | 3.9–12 | 6.0 (2.1) |
| Hgb (g/dl) | 13.2–18 | 10.5 (1.5) * |
| Lymphocytes (k/ul) | 1.3–4.2 | 1.5 (0.7) |
| Neutrophils (k/ul) | 1.3–7.5 | 3.7 (1.5) |
| Monocytes (k/ul) | 0.4–1 | 0.5 (0.2) |
| Eosinophils (k/ul) | 0.2–0.5 | 0.2 (0.2) |

*Indicates value outside of normal range.

# Results
## Demographic and clinical characterization

Demographic and clinical data of the 20 maintenance hemodialysis (HD) and controls (HC) are summarized in *Table 1*. The racial distribution differed between cohorts with more Black/ African American subjects in the HD cohort. The cohorts were otherwise demographically similar. The subjects within the HD cohort had significantly more comorbidities, most notable of which include type 2 diabetes mellitus (T2DM), hypertension (HTN), dyslipidemia, and other cardiovascular conditions. The most common causes of renal failure were T2DM and HTN, with a minority of cases attributed to anatomic defects (reflux uropathy) and autoimmune conditions (systemic lupus erythematous and idiopathic thrombocytopenic purpura).

There were eight HD subjects who previously tested positive for SARS-CoV-2, with positive test dates ranging from 7 mo to 4 wk preceding vaccination. Five control subjects self-reported a prior positive SARS-CoV-2 test, with positive test dates ranging from 8 mo to 4 wk preceding vaccination. Detailed clinical characterization of HD subjects is summarized in *Table 2*. Notable laboratory data includes an elevated ferritin from normal (with high population variance) and anemia.

All subjects received two BTN162b2 vaccination doses with the second dose (V2) administered 3 wk after the first (V1). Anti-spike IgG binding and neutralizing assay data were obtained for all subjects prior to V1 (V1D0) and 7 d after V2 (V2D7). RNA sequencing data was obtained for all control subjects prior to each vaccination dose (D0), and at 1 d (D1) and 7 d (D7) after each dose, corresponding to six time points: V1D0, V1D1, V1D7, V2D0, V2D1, and V2D7. One control subject is missing V2D0 data, and one is missing V2D1 data. RNA sequencing data was obtained for 12 HD subjects prior to each vaccination dose, and at 2 d (D2) and 7 d after each dose, corresponding to six time points: V1D0, V1D2, V1D7, V2D0, V2D2, and V2D7. Two HD subjects are missing V2D2 data. Sequencing data was not obtained for subjects with fewer than five time points of successfully constructed RNA libraries due to time points without sample collection or failure to extract high-quality mRNA from PBMCs. Six-month follow-up (M6) anti-spike IgG binding titers were obtained for 15 HC subjects and 19 HD subjects. One HD subject tested positive for SARS-CoV-2 14 d after the second vaccination dose, demonstrating mild symptoms. None of the other subjects reported SARS-CoV-2 infection up to 6 mo follow-up after the second vaccination.

## Differential gene expression analysis

To characterize the molecular basis of immune responses to vaccination in HC and HD, we performed differential gene expression analyses of the PBMC RNA sequencing data. There are substantially more differentially expressed genes (DEGs) in response to V2 compared to V1, and at D1 and D2 postvaccination compared to D7 (*Figure 1—figure supplement 1*). For HC, the largest number of DEGs is found at V2D1 (3974), indicating the most transcriptional activity immediately after the second vaccine dose, followed by V2D7 (177), V1D1 (128), and V1D7 (10). HD follows a similar pattern, with the largest number of DEGs found at V2D2 (1,111), followed by V2D7 (153), V1D2 (68), and V1D7 (8). Notably, HD subjects with no SARS-CoV-2 history (n = 6) have substantially lower numbers of DEGs than HD subjects with positive SARS-CoV-2 history (n = 6) at each time point, and particularly at V2 time points (*Figure 1A*).Direct comparison of gene expression between HC and HD with no prior reported SARS-CoV-2 infection at V1D7 yielded five DEGs in HD versus HC including increased expression of

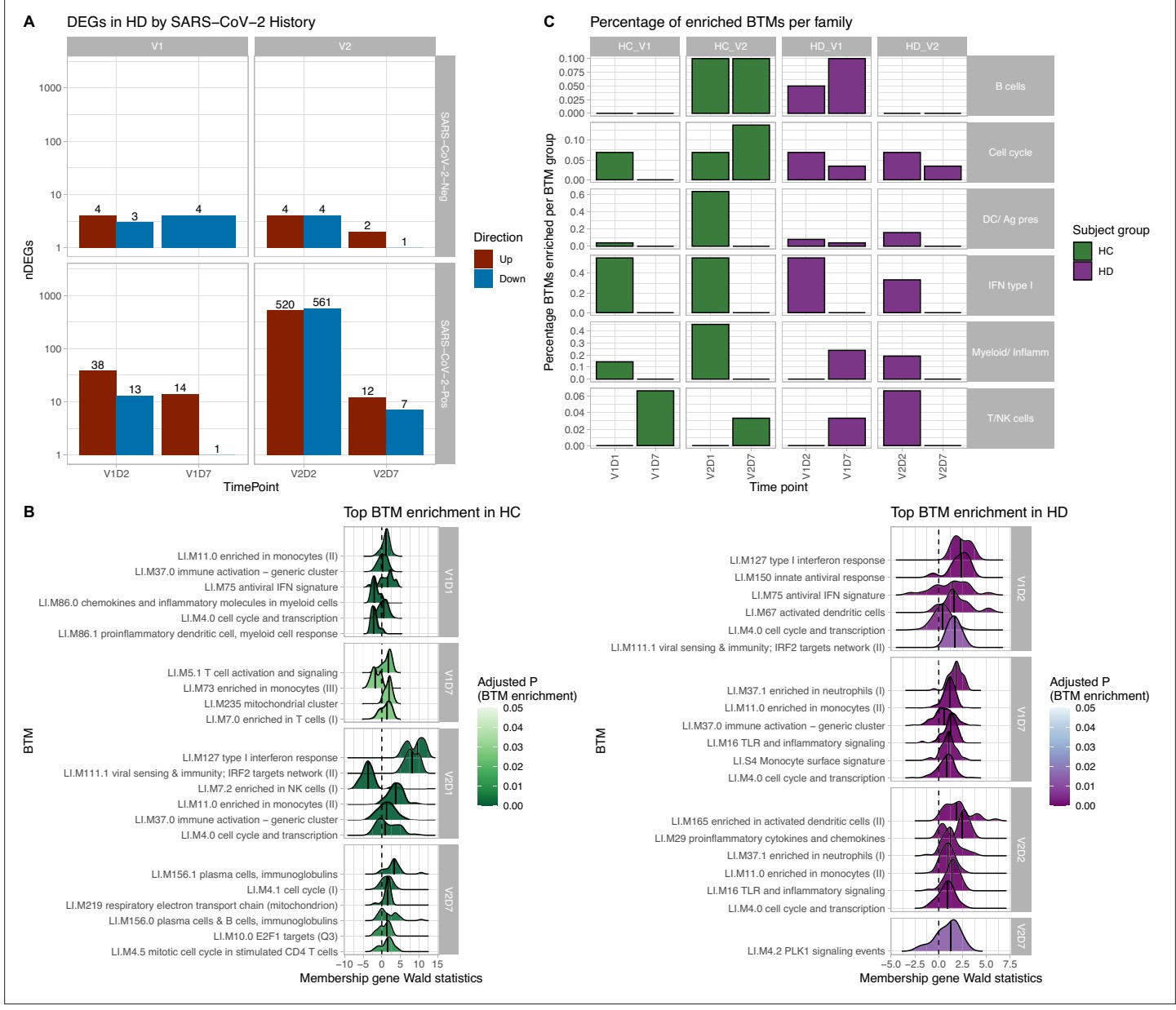

**Figure 1.** Results of differential gene expression and blood transcription module (BTM) analysis. (**A**) Differentially expressed genes (DEGs) increased after second vaccination dose compared to first, and at early time points compared to day 7 (D7). DEGs are shown for maintenance hemodialysis (HD) with (n = 6) and without (n = 6) prior SARS-CoV-2 history. See *Figure 1—figure supplement 1* for healthy control (HC) and HD data independent of SARS-CoV-2 history. The DESeq2 R package was used to identify genes that were differentially expressed at each time point after vaccination for each subject group (p<0.05, FDR-adjusted). (**B**) HC and HD with no SARS-CoV-2 history demonstrate distinct longitudinal enrichments of BTMs. Left: in HC, the most significantly enriched BTMs are shown (up to six) for day 1 (D1) and day 7 (D7) after each vaccination dose (V1, V2) (p<0.05, FDR-adjusted). Density plots for each BTM represent Wald statistics from DESeq2 analysis for each membership gene, thereby representing increased or decreased expression per gene at each time point compared to baseline (V1D0 or V2D0). Right: similarly in HD, the most significantly enriched BTMs for day 2 (D2) and day 7 (D7) are shown. (**C**) HD demonstrates diminished transitioning from innate to adaptive immune BTM enrichment after each vaccination dose. Percentages of BTMs in each BTM family with significant enrichment are shown at each time point after each vaccination dose (V1, V2) for day 1 (D1) and day 7 (D7) in HC, and day 2 (D2) and D7 for HD in subjects with no prior infection with SARS-CoV-2 (p<0.05, FDR-adjusted). Direction of enrichment was determined using the median Wald statistic from DESeq2 analysis for each BTM membership gene, thereby representing overall increased or decreased expression of membership genes at each time point compared to baseline (V1D0 or V2D0).

The online version of this article includes the following figure supplement(s) for figure 1:

**Figure supplement 1.** Differentially expressed genes (DEGs) increased after second vaccination dose compared to first, and at day 1 (D1, for healthy controls [HC]) and day 2 (D2, for maintenance hemodialysis [HD] compared to day 7 [D7]).

*Figure 1 continued on next page*

*Figure 1 continued*

**Figure supplement 2.** Hemodialysis (HD) patients without prior SARS-CoV-2 infection show increased myeloid activity at V1D7 and decreased metabolic activity at V2D7 compared to controls (HC).

**Figure supplement 3.** Hemodialysis (HD) patients with prior SARS-CoV-2 infection show increased expression of innate and adaptive immune blood transcription modules (BTMs) post-vaccination.

chemokine CCL19 in HD ($p<0.05$, FDR-corrected). Comparison of these same groups at V2D7 yielded 18 DEGs including increased expression in HD of TIA1, which encodes a granule-associated protein expressed in cytolytic lymphocytes (*Anderson et al., 1990*) and natural killer cells, and BH3, a pro-apoptotic Bcl-2 family member and mediator of lymphocyte apoptosis (*Labi et al., 2008*).

## Blood transcription module (BTM) enrichment

BTM enrichment analysis of subjects without SARS-CoV-2 history reveals the vaccine-induced progression of various immune processes at each time point after vaccination (*Figure 1B*). Following V1, HC demonstrate early (V1D1) enrichment of 29 BTMs, with substantial upregulation of monocyte and antiviral IFN activity (*Supplementary file 1*). The immune response transitions to V1D7 enrichment of four BTMs including significant T cell activation and downregulation of monocytes (*Supplementary file 2*). Following V2, HC demonstrate early (V2D1) enrichment of 82 BTMs, with substantial upregulation of innate antiviral activity, similarly to V1D1 (*Supplementary file 3*). The immune response transitions to V2D7 enrichment of ten BTMs, with significant upregulation of plasma cells and immunoglobulins (*Supplementary file 4*).

In contrast, HD demonstrate early (V1D2) enrichment of 12 BTMs after the first vaccination dose, most significantly involving upregulation of innate antiviral responses (*Figure 1B*, *Supplementary file 5*). The immune response transitions to V1D7 enrichment of 17 BTMs, with substantial upregulation of myeloid modules (*Figure 1B*, *Supplementary file 6*). The V1D7-positive enrichment of monocyte/myeloid modules in HD contrast the negative enrichment of these modules in HC (*Figure 1B*, *Figure 1—figure supplement 2*). Following the second vaccination dose, HD demonstrate early (V2D2) enrichment of 27 BTMs most significantly involving upregulation of dendritic cell activity and proinflammatory cytokines and chemokines (*Figure 1C*, *Supplementary file 7*). The immune response progresses to V2D7 enrichment of one BTM: PLK signaling events (*Figure 1B*).

While there were no significant BTM enrichments in HC with positive SARS-CoV-2 history, most likely due to the insufficient number of subjects, BTM enrichments for HD with positive SARS-CoV-2 demonstrated notable upregulation of plasma cell activity at V1D7. This contrasts with V1D7 for HD with negative SARS-CoV-2, which show primary enrichment of myeloid BTMs (*Figure 1B*). The remainder of these enrichments are shown in *Figure 1—figure supplement 3*.

Summary enrichments using BTM families show many positive early V1 enrichments of type 1 IFN activity that dissipate by V1D7 in both HC and HD (*Figure 1C*). However, HC show early positive and negative enrichments of myeloid/inflammatory family activity that dissipate by V1D7, while HD show many early positive enrichments of myeloid/inflammatory family activity that persist and increase at V1D7. Following V2, HC show early predominance of dendritic cell (DC)/antigen presenting cell (APC), IFN type I, and myeloid/inflammatory family activity transitioning to B cell and cell cycle activity at V2D7, while HD show predominant early IFN type I family activity transitioning to just one detectable cell cycle module enrichment.

## Antibody binding and neutralization assay response

We next aimed to assess immune protection conferred by the vaccine through quantification of anti-spike IgG antibodies and functional assessment of neutralizing antibodies. All subjects demonstrated an increase in anti-spike IgG at V2D7, with titers for all subjects except one still elevated above baseline at six months. The exception was one HD subject with prior SARS-CoV-2 infection who demonstrated the highest baseline titers of all subjects prior to vaccination. Both HC and HD subjects demonstrated a statistically significant increase in anti-spike IgG and neutralization activity (ID50) from V1D0 to V2D7 ($p<0.001$), followed by an expected decrease at M6 from V2D7 levels ($p<0.001$) (*Figure 2*). Despite this decrease, M6 titers were still increased compared to baseline ($p<0.001$).

Higher anti-spike IgG at V2D7 was significantly predicted by higher pre-vaccination anti-spike IgG, control group assignment, and younger age ($p<0.01$, $p<0.05$, $p<0.05$, respectively), while gender,

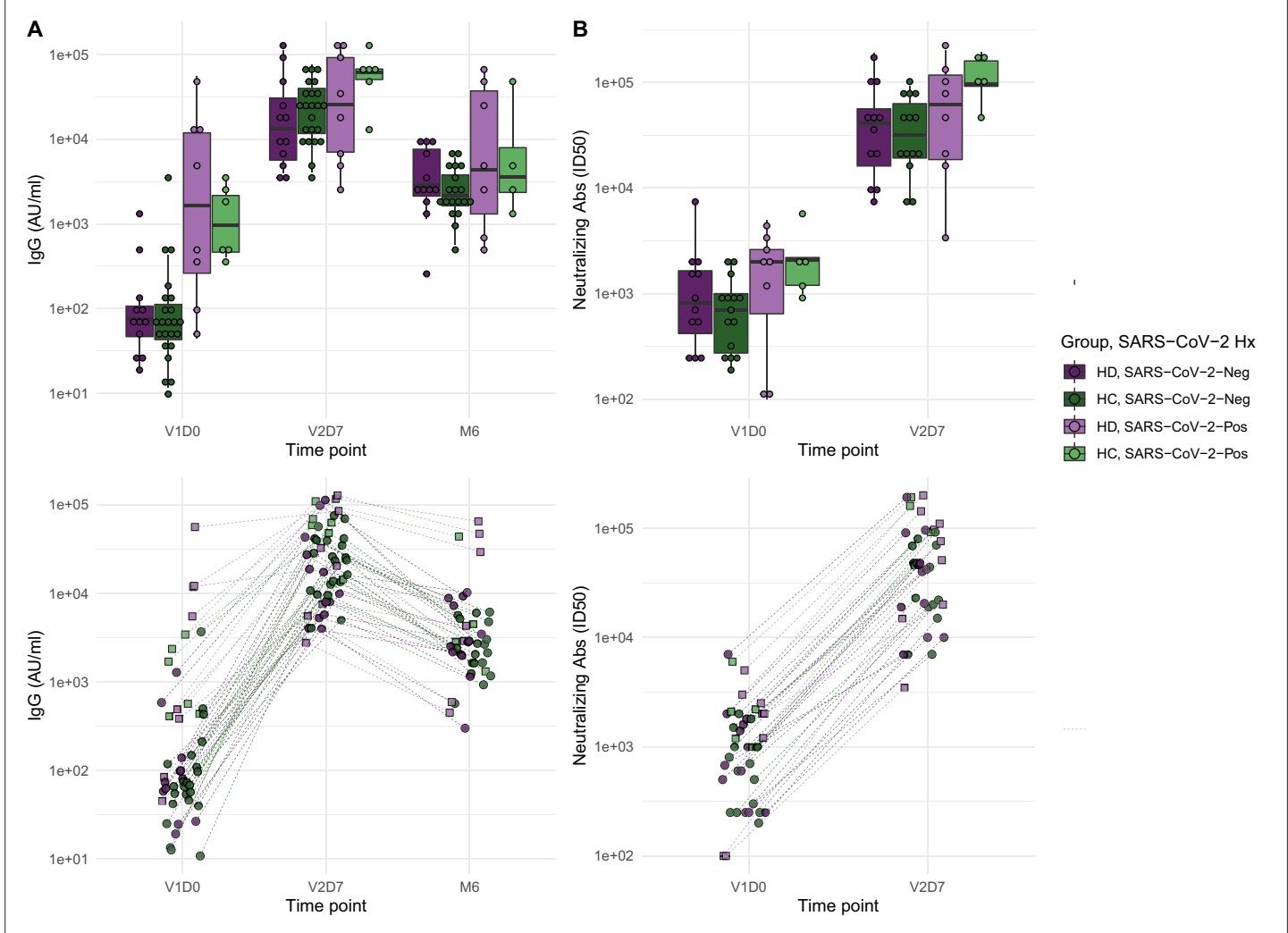

**Figure 2.** Antibodies significantly increased in both healthy control (HC) and maintenance hemodialysis (HD) groups 1 wk after the second vaccination dose (p<0.001) and 6 mo after initial vaccination (p<0.001) with the BNT162b2 mRNA COVID-19 vaccine. (**A**) Anti-spike IgG levels in HC and HD subjects with and without prior SARS-CoV-2 history before vaccination (V1D0), 1 wk after second vaccination dose (V2D7), and 6 mo after initial vaccination (M6). Higher anti-spike IgG at V2D7 was significantly predicted by higher pre-vaccination anti-spike IgG, being a member of HC, and younger age (p<0.01, p<0.05, p<0.05, respectively). Higher anti-spike IgG at M6 was significantly predicted by higher V2D7 anti-spike IgG (p<0.001). (**B**) Antibody neutralization activity (ID50) in controls and HD subjects with and without prior SARS-CoV-2 history at V1D0 and V2D7. Higher neutralization activity (ID50) at V2D7 was significantly predicted by higher pre-vaccination ID50, with no additional predictive value conferred by subject group.

race, and ethnicity were not. Higher anti-spike IgG at M6 was significantly predicted by higher V2D7 anti-spike IgG (p<0.001), with no additional predictive value conferred by SARS-CoV-2 history, subject group, age, gender, race, or ethnicity. Higher neutralization activity (ID50) at V2D7 was significantly predicted by higher pre-vaccination ID50, with no additional predictive value conferred by subject group, age, gender, race, and ethnicity.

## Transcriptomic and clinical predictors of antibody binding response in HD

Linear models to predict anti-Spike IgG at V2D7 and at M6 in HD using enriched BTMs, controlling for SARS-CoV-2 history, identified BTM predictors at all time points except for V1D2. Of the 18 enriched BTMs at V1D7, increased expression (from V1D0) of 'LI.M156.1 plasma cells, immunoglobulins' was predictive of higher anti-spike IgG at V2D7 (p<0.05, FDR-corrected), controlling for SARS-CoV-2 history. Of the 30 enriched BTMs at V2D2, increased expression of 18 BTMs was predictive of higher anti-spike IgG at V2D7 (p<0.05, FDR-corrected). These include innate immune, antigen presentation,

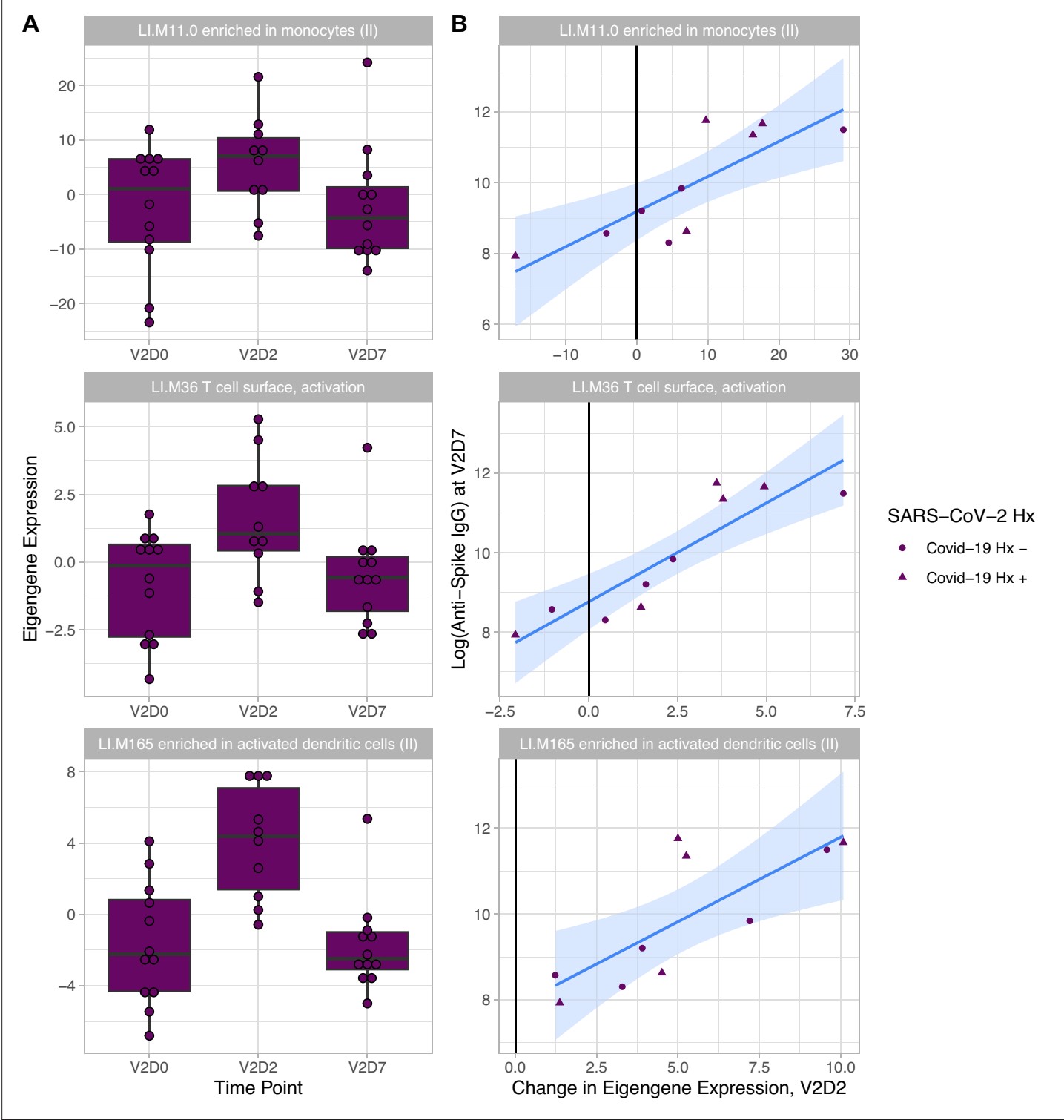

**Figure 3.** Increased expression of multiple blood transcription modules (BTMs) at V2D2 is predictive of higher anti-spike IgG at V2D7. Of 30 enriched BTMs at V2D2, increased expression of 18 BTMs is predictive of increased anti-spike IgG at V2D7 (p<0.05, FDR-corrected), controlling for SARS-CoV-2 history. Predictive pathways include innate immune, antigen presentation, and T cell pathways. The eigengenes (**A**) and linear models (**B**) for three example BTMs are shown.

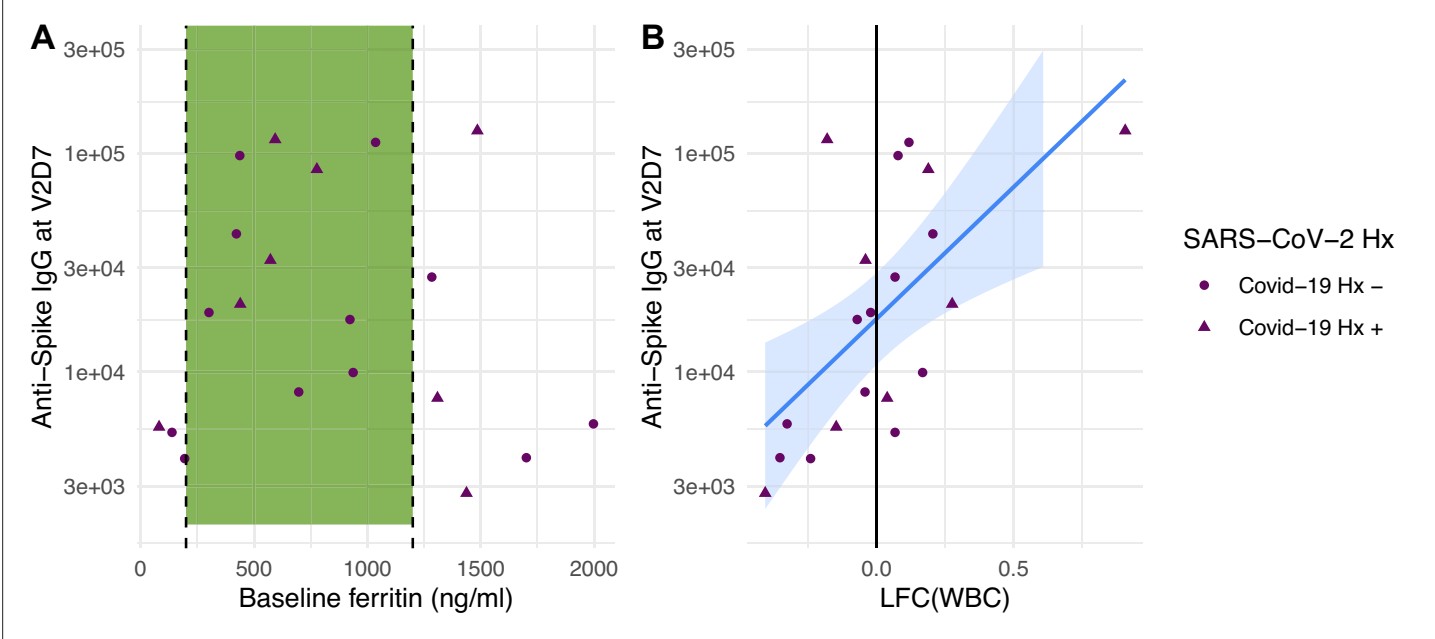

**Figure 4.** Baseline ferritin level and post-V1 white blood cell count (WBC) are clinical predictors of post-V2 antibody responses in maintenance hemodialysis (HD) patients. (**A**) Ferritin levels associated with lowest all-cause mortality predict the development of higher anti-spike IgG after vaccination at V2D7 (p<0.01) and M6 (not shown, p<0.05) in maintenance HD patients. Dashed vertical lines indicate the intermediate range of ferritin (200–1200 ng/ml) associated with lowest all-cause mortality (*Kalantar-Zadeh et al., 2005*). (**B**) Increased WBC after first vaccination dose is predictive of anti-spike IgG titers after vaccination at V2D7 (p<0.01) and M6 (not shown, p<0.05) in maintenance HD patients. Points with negative log-fold change of white blood cell counts (LFC(WBC)) and positive LFC(WBC) represent a decrease and increase, respectively, in WBC from baseline labs.

and T cell pathways (*Figure 3*). Increased expression of 'LI.M4.2 PLK1 signaling events' at V2D7 compared to V2D0, which was the only enriched module at this time point for HD subjects with no SARS-CoV-2 history, was predictive of higher anti-spike IgG at both V2D7 and M6 (p<0.05).

Linear models to predict anti-spike IgG at V2D7 and at M6 In HD using clinical predictors yielded significant baseline and post-vaccination predictors. Baseline ferritin levels in the intermediate range (200–2000 ng/ml) were associated with higher anti-spike IgG at V2D7 and M6 (p<0.01, p<0.05), controlling for history of SARS-CoV-2. URR, WBC counts, transferrin saturation, and hemoglobin were not significant predictors of antibody development. *Figure 4A* shows anti-spike IgG at V2D7 as a function of baseline ferritin levels, identifying the intermediate range of ferritin which has previously been associated with lowest all-cause mortality (*Kalantar-Zadeh et al., 2005*).

The LFC of WBCs from baseline after the first vaccination dose was significantly predictive of antibody titer levels at both V2D7 (p<0.01) and M6 (p<0.05), controlling for SARS-CoV-2 history and number of days after vaccination that labs were collected (*Figure 4B*). The predictive value of LFC of WBCs is predominantly driven by increased lymphocyte counts; LFC of absolute lymphocyte counts was predictive of V2D7 (p<0.01) and M6 (trend-level, p=0.056) antibody titers, controlling for initial antibody titers and date of clinical labs.

## Discussion

Our results demonstrate differing expression of BTMs and differing time courses of immune responses to the BTN162b2 mRNA COVID-19 vaccination in maintenance HD subjects compared to controls. Controls demonstrated expected transitions from early type I interferon and myeloid activity to T cell activity after the first vaccination dose (*Figure 1*). The predominant positive enrichment of T cell modules in controls at 1 wk after the first vaccination dose (V1D7) was contrasted with predominant positive enrichment of myeloid modules in HD at V1D7. Interestingly, HD showed prolonged upregulation of myeloid activity at V1D7, while controls showed downregulation of these modules at V1D7 (*Figure 1B*, *Figure 1—figure supplement 2*). Overall, these observations indicate delayed

resolution of innate myeloid responses in the HD cohort and suggest diminished transition to an adaptive immune response.

Following the second vaccination dose, both groups demonstrated early enrichment of innate immune modules, with HC alone transitioning to a plasma cell response by V2D7 (*Figure 1C*). Direct group comparisons at V2D7 did not show differences of plasma cell response, but metabolic activity was decreased in HD compared to controls (*Figure 1—figure supplement 2*). Further, HD demonstrated increased V2D7 expression compared to controls of pro-apoptotic Bcl-2 family member BH3, a mediator of lymphocyte apoptosis. A prior study showed accelerated in vitro apoptosis of lymphocytes in uremia, with a particularly pronounced effect on B cells, mediated by dysregulation of Bcl-2. These results suggest a state of heightened cellular stress in HD after vaccination leading to increased apoptotic signaling.

Despite differing transcriptomic time courses in the two group, our results demonstrate significant elevation of anti-spike IgG titers after two doses of BNT162b2 mRNA COVID-19 vaccination in both HD and controls. HD demonstrated only a slight decrease of IgG levels at V2D7 when controlling for SARS-CoV-2 history ($p<0.05$) and no statistically significant difference at 6 mo. Prior studies comparing short-term antibody response to BNT162b2 mRNA COVID-19 vaccination in HD versus controls find antibody response rates of 84–96% in HD after two vaccination doses, but with lower mean IgG levels compared to controls (*Grupper et al., 2021*; *Jahn et al., 2021*; *Attias et al., 2021*; *Anand et al., 2021*; *Agur et al., 2021*; *Longlune et al., 2021*; *Drakesmith et al., 2021*). Notably, the HD population studied here is younger and more racially and ethnically diverse. The average age of HD cohorts in prior studies was predominantly in the 60s, compared to an average age of 54 in our study. Jahn et al. found in a subset analysis that HD patients under 60 y of age responded equally to healthy controls, suggesting an interaction between increasing age and less effective antibody response in HD patients (*Jahn et al., 2021*).

HD subjects with documented SARS-CoV-2 infection prior to vaccination had wider variance of antibody titers at all time points in this study, with two subjects demonstrating V1D0 antibody titer levels similar to that of uninfected subjects. These two subjects consistently had the lowest titer levels at V2D7 and M6 within the group of previously infected subjects and amongst the lowest titers across all subjects. One subject is the oldest enrolled patient, and both are diagnosed with hyperlipidemia.

Given previously and presently demonstrated the wider variance of protective immune responses in HD and altered interactions with risk factors including age, it is valuable to identify predictors of the strength of immune response to vaccination in this population. We identified both transcriptomic and clinical predictors of anti-spike IgG development at both V2D7 and 6 mo after the second vaccination dose (M6). Increased gene expression of blood transcription modules (BTMs) including monocyte activity, dendritic cell and antigen presentation activity, IFN type I activity, and T cell activation 2 d after the second vaccination dose (V2D2) in HD were predictive of V2D7 anti-spike IgG. Additionally, increased expression of PLK1 signaling events, indicating increased cell cycle activity, at V2D7 was predictive of V2D7 and M6 anti-spike IgG. Clinically, serum ferritin values in the intermediate range at baseline predicted stronger anti-spike IgG development. A prior study of 58,058 maintenance HD subjects found serum ferritin levels between 200 and 1200 ng/ml to be associated with lower all-cause mortality due to ferritin <200 ng/ml representing low iron status, and >1200 ng/ml representing a hyper-inflammatory state due to ferritin's status as an acute phase reactant (*Kalantar-Zadeh et al., 2005*). Iron deficiency has been linked to impaired immune response and vaccine efficacy in other infections, while inflammation induces macrophage release of the heavy chain component of ferritin, FTH, which has been reported to inhibit lymphocyte proliferation and function (*Drakesmith et al., 2021*; *Kernan and Carcillo, 2017*). Additionally, increased LFC in WBC count 1–3 wk after vaccination was predictive of higher antibody titers.

Our study is limited by different early time points between controls and HD (day 1 vs. day 2 after each vaccination dose), sample size particularly when subdividing SARS-CoV-2 history, and differing racial profiles of the control and HD cohort. The smaller sample size additionally limits our ability to characterize differential immune pathways in clinical subsets of the dialysis population, such as those with low, medium, and high baseline ferritin levels. Our small sample size precludes a validation cohort to validate the study's results. We also cannot exclude the likelihood that these results are driven in part by immune dysregulation from diabetes mellitus, hypertension, and/or hyperlipidemia. While we did not exclude these in recruitment of our control cohort,

subjects in the HD cohort are inevitably disproportionately affected by these conditions. Future studies are needed for more comprehensive characterization of the immune pathway recruitment in response to the COVID-19 vaccinations in this population. It will be of further interest to characterize immune responses dependent on etiology of ESRD (i.e., hypertension vs. diabetes mellitus), as well as amount of time spent on hemodialysis, as this study was not sufficiently powered to disentangle these differences.

The COVID-19 mRNA vaccines have proven more efficacious than other vaccines in the ESRD population, both for COVID-19 and for other infectious diseases; for example, there has been an inconsistent antibody response in HD to the adenovirus vector-based COVID-19 vaccine Ad26. COV2.S compared to mRNA vaccines (*Brunelli et al., 2022*). Additionally, while more than 90% of patients without chronic kidney disease develop protective antibodies against hepatitis B after vaccination, only 50–60% of patients with ESRD seroconvert (*Eleftheriadis et al., 2007*). One explanation is that, in mRNA vaccines, the mRNA both encodes the viral antigen and acts as an adjuvant due to its innate immunostimulatory properties; the mRNA is recognized by endosomal and cytosolic innate sensors upon cell entry, resulting in cellular activation and production of type I interferons and other inflammatory mediators (*Teijaro and Farber, 2021*). This elevated innate immune stimulus could overcome immune desensitization in ESRD, evidenced by diminished TLR4 expression on monocytes (*Koc et al., 2011*) and downregulation of activating receptors on natural killer cells in this population (*Nagai, 2021*). If so, the mRNA vaccine delivery vehicle could prove particularly valuable in vaccine development for ESRD and HD going forward.

Overall, we demonstrate differing time courses of immune responses to the BTN162b2 mRNA COVID-19 vaccination in maintenance HD subjects and identify transcriptomic and clinical predictors of anti-spike IgG titers in HD. Given the efficacy of this vaccination in our HD cohort, the differential transcriptomic responses likely represent immune alterations with subclinical effects on humoral immune response to the BTN162b2 mRNA COVID-19 vaccine. Our results warrant further characterization of the immune dysregulation of ESRD and of immune biomarkers that underlie efficacious immune responses to vaccination in this population.

## Acknowledgements

This research was funded by the University of Illinois at Chicago Center for Clinical and Translational Science (CCTS) award UL1TR002003.

## Additional information

### Competing interests

Richard M Novak: Received a grant from Janssen. The author has received consulting fees from Gilead and Viiv. The author has no other competing interests to declare. The other authors declare that no competing interests exist.

### Funding

| Funder | Grant reference number | Author |
| --- | --- | --- |
| Eunice Kennedy Shriver National Institute of Child Health and Human Development | F30HD102093 | Yi-Shin Chang |
| Division of Intramural Research | F30HL151182 | Kai Huang |
| Division of Intramural Research | T32HL144909 | Christen L Vagts |
| Division of Intramural Research | R01HL138628 | Patricia W Finn David L Perkins |

| Funder | Grant reference number | Author |
| --- | --- | --- |
| Center for Clinical and Translational Science, University of Illinois at Chicago | UL1TR002003 | David L Perkins |

The funders had no role in study design, data collection and interpretation, or the decision to submit the work for publication.

## Author contributions

Yi-Shin Chang, Kai Huang, Conceptualization, Data curation, Formal analysis, Funding acquisition, Validation, Investigation, Visualization, Methodology, Writing - original draft, Writing – review and editing; Jessica M Lee, Conceptualization, Formal analysis, Investigation, Writing – review and editing; Christen L Vagts, Christian Ascoli, Claudia M Lora, Conceptualization, Writing – review and editing; Md-Ruhul Amin, Mahmood Ghassemi, Russell Edafetanure-Ibeh, Yue Huang, Ruth A Cherian, Nandini Sarup, Samantha R Warpecha, Sunghyun Hwang, Rhea Goel, Montserrat Hernandez, Yang Chen, Juli-anne Jorgensen, Wangfei Wang, Mladen Rasic, Investigation, Writing – review and editing; Benjamin A Turturice, Cody Schott, Conceptualization, Investigation, Writing – review and editing; Richard M Novak, Resources, Methodology, Writing – review and editing; Patricia W Finn, David L Perkins, Conceptualization, Resources, Supervision, Funding acquisition, Project administration, Writing – review and editing

## Author ORCIDs

Yi-Shin Chang ⓘ http://orcid.org/0000-0002-9539-4553
Kai Huang ⓘ http://orcid.org/0000-0003-2445-2696
Jessica M Lee ⓘ https://orcid.org/0000-0002-4492-2221
Christian Ascoli ⓘ http://orcid.org/0000-0002-7424-2710
Benjamin A Turturice ⓘ https://orcid.org/0000-0001-9382-4612
Patricia W Finn ⓘ https://orcid.org/0000-0001-6936-103X

## Ethics

Human subjects: The study was approved by the University of Illinois at Chicago IRB (#2018-1038) Ethics Review Committee. Maintenance HD patients undergoing vaccination with the BNT162b2 mRNA COVID-19 vaccine in February 2021 were recruited from the outpatient HD unit at the University of Illinois Hospital (UIH) in Chicago, IL. Control subjects consisted of UIH employees undergoing BNT162b2 mRNA COVID-19 vaccination at UIH from December 2020 to January 2021 with no self-reported history of kidney disease or immune disorders. All subjects provided informed consent for participation in this research and for publication of results.

## Decision letter and Author response

Decision letter https://doi.org/10.7554/eLife.83641.sa1
Author response https://doi.org/10.7554/eLife.83641.sa2

---

# Additional files

## Supplementary files

• Supplementary file 1. Significantly enriched blood transcription modules at day 1 in healthy controls (HC) following the first vaccination dose (V1D1).

• Supplementary file 2. Significantly enriched blood transcription modules at day 7 in healthy controls (HC) following the first vaccination dose (V1D7).

• Supplementary file 3. Significantly enriched blood transcription modules at day 1 in healthy controls (HC) following the second vaccination dose (V2D1).

• Supplementary file 4. Significantly enriched blood transcription modules at day 7 in healthy controls (HC) following the second vaccination dose (V2D7).

• Supplementary file 5. Significantly enriched blood transcription modules at day 2 in hemodialysis subjects (HD) following the first vaccination dose (V1D2).

• Supplementary file 6. Significantly enriched blood transcription modules at day 7 in hemodialysis

subjects (HD) following the first vaccination dose (V1D7).

• Supplementary file 7. Significantly enriched blood transcription modules at day 2 in hemodialysis subjects (HD) following the second vaccination dose (V2D2).

• MDAR checklist

### Data availability

Data Availability: Raw FASTQ files from bulk RNA-seq of all subjects in this dataset have been deposited in GEO under accession code GSE209985. All data were generated using the NovaSeq 6000 platform. Datasets Generated: bulk RNA-seq data. Reporting Standards: N/A.

The following dataset was generated:

| Author(s) | Year | Dataset title | Dataset URL | Database and Identifier |
|---|---|---|---|---|
| Huang K, Chang Y, Lee JM, Vagts CL, Ascoli C, Amin M, Ghassemi M, Lora CM, Edafetanure-Ibeh R, Huang Y, Cherian RA, Sarup N, Warpecha SR, Hwang S, Goel R, Turturice BA, Schott C, Hernandez M, Chen Y, Jorgensen J, Wang W, Rasic M, Novak RM, Finn PW, Perkins DL | 2024 | Response to mRNA vaccine BNT162b2 in hemodialysis patients | https://www.ncbi.nlm.nih.gov/geo/query/acc.cgi?acc=GSE209985 | NCBI Gene Expression Omnibus, GSE209985 |

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
