## [Editor Report]

Chang et al. demonstrate through their findings that COVID-19 mRNA vaccination of hemodialysis patients produces no significant difference in antibody levels achieved across the vaccination series. However, they find that T-cell responses may be delayed in hemodialysis patients as they have lower activation of T-cell genes than healthy controls. The RNA sequencing evidence is solid. However, they lack data on a clinical correlation to T-cell responses.

---

## [Decision Letter]

**Decision letter after peer review:**

Thank you for submitting your article "Immune response to the mRNA COVID-19 vaccine in hemodialysis patients: cohort study" for consideration by *eLife*. Your article has been reviewed by two peer reviewers, one of whom is a member of our Board of Reviewing Editors, and the evaluation has been overseen by Mone Zaidi, as the Senior Editor, have overseen the evaluation. The reviewers have opted to remain anonymous.

The Reviewing Editor has drafted this to help you prepare a revised submission.

Essential revisions (for the authors):

1) Please address reviewer two's comments about the cohort study groups' shortcomings. Additional data is needed to justify the conclusions.

2) Please address the manuscript presentation shortcomings identified by reviewer one.

*Reviewer #1 (Recommendations for the authors):*

While the concept of the article is interesting, the data is poorly presented. The figures feel like more something in a dissertation than a scientific paper in that there is an overabundance of data presentation that does not necessarily drive differential conclusions or advance the scientific question. For example, I would suggest summarizing figures 1-3 with one more easily understood and comprehensive figure.

In Figure 4, the figure in my mind concludes that dialysis patients have a statistically similar response to controls. That point is lost in the results and the text of the figure. Moreover, the legend, coloring scheme, and naming could be improved to enhance readability.

The exciting finding that T-cell signatures are muted in dialysis patients does not seem clinically supported by any data-ideally, the collection, and analysis of T-cell responses in dialysis patients and controls.

The scientific premise in the discussion was that mRNA might be a superior vaccination methodology compared to traditional protein vaccines. This supposition would be significantly strengthened if there was some comparison made for dialysis patients of more traditional vaccines protein-based like the Ortho vaccine compared to mRNA vaccines in dialysis patients. Please include that in the discussion if data are available from published sources.

*Reviewer #2 (Recommendations for the authors):*

To strengthen this manuscript, the reviewer has some suggestions below:

1. For publication in *eLife* Journal, a bigger sample size of both cohorts and matching the number of candidates of different races would be appreciated, although the authors described this weakness in the MS.

2. Setting up a validation cohort, either internal or external. Add data or discuss this limitation.

3. Stratifying DM/hypertension/Dyslipidemia patients to compare with those without metabolic diseases in the HD cohort and see how metabolic diseases cause different immune responses in HD patients.

4. Do these immune response differences also related to the years and the styles of HD?

5. The DM and hypertension candidates in the control cohorts could be deleted. Or the authors need to discuss the limitations in terms of comparison.

---

## [Author Response]

Essential revisions (for the authors):Reviewer #1 (Recommendations for the authors):While the concept of the article is interesting, the data is poorly presented. The figures feel like more something in a dissertation than a scientific paper in that there is an overabundance of data presentation that does not necessarily drive differential conclusions or advance the scientific question. For example, I would suggest summarizing figures 1-3 with one more easily understood and comprehensive figure.

We appreciate the reviewer’s comments. We have combined original figures 1,2,3 into Figure 1 (A, B, and C). From original figure 1, we have also eliminated most of the figure panes, only retaining the numbers of differentially expressed genes at each time point for hemodialysis with and without SARS-CoV-2 history. The data that has been removed from original figure 1 is now summarized in the text and included as supplemental figure S1.

“For HC, the largest number of DEGs is found at V2D1 (3974), indicating the most transcriptional activity immediately after the 2^nd^ vaccine dose, followed by V2D7 (177), V1D1 (128), and V1D7 (10). HD follows a similar pattern, with the largest number of DEGs found at V2D2 (1111), followed by V2D7 (153), V1D2 (68), and V1D7 (8).” We have also removed extraneous data from original Figure 3 and presented the data in a more digestible format (as Figure 1C). We have also reworded parts of the new figure legend for Figure 1A-C to summarize conclusions more clearly.”

In Figure 4, the figure in my mind concludes that dialysis patients have a statistically similar response to controls. That point is lost in the results and the text of the figure. Moreover, the legend, coloring scheme, and naming could be improved to enhance readability.

We have changed the coloring scheme, legend, and subject naming to enhance readability of this figure (now Figure 2). While both groups did mount a significant antibody response, we did find that membership in the control group was a predictor of anti-spike IgG at V2D7. We have included these statistical results in the legend (in addition to its presence in the text) to clarify these results.

The exciting finding that T-cell signatures are muted in dialysis patients does not seem clinically supported by any data-ideally, the collection, and analysis of T-cell responses in dialysis patients and controls.

We note that the control subject transcriptomic responses are dominated by T and B-cell responses by Day 7 after both the first and second day of vaccination, while HD subject transcriptomic responses continue to be dominated by innate, and in particular myeloid, immune signatures. However, the reviewer is correct that direct comparison of adaptive immune responses at D7 time points between controls and hemodialysis do not reach statistical significance.

We have revised the Results section in our abstract to state: “Transcriptomic analyses demonstrated differing time courses of immune responses, with prolonged myeloid cell activity in HD at one week after the first vaccination dose. HD also demonstrated decreased metabolic activity and decreased antigen presentation compared to controls after the second vaccination dose.” In our discussion, we state “Overall, these observations indicate delayed resolution of innate myeloid responses in the HD cohort, and suggest diminished transition to an adaptive immune response.”

The scientific premise in the discussion was that mRNA might be a superior vaccination methodology compared to traditional protein vaccines. This supposition would be significantly strengthened if there was some comparison made for dialysis patients of more traditional vaccines protein-based like the Ortho vaccine compared to mRNA vaccines in dialysis patients. Please include that in the discussion if data are available from published sources.

We have added a reference to the following study which shows inconsistent antibody response in HD to the Johnson and Johnson adenovirus-based Covid-19 vaccine in comparison to mRNA Covid-19 vaccines:

Brunelli SM, Sibbel S, Karpinski S, et al. Comparative Effectiveness of mRNA-based BNT162b2 Vaccine versus Adenovirus Vector–Based Ad26.COV2.S Vaccine for the Prevention of COVID-19 among Dialysis Patients. *JASN*. 33(4):688-697. doi: 10.1681/ASN.2021101395

The following text has been added to the discussion:

“The Covid-19 mRNA vaccines have proven more efficacious than other vaccines in the ESRD population, both for Covid-19 as well as for other infectious diseases; for example, there has been an inconsistent antibody response in HD to the adenovirus vector-based Covid-19 vaccine Ad26.COV2.S compared to mRNA vaccines ^36^. Additionally, while more than 90% of patients without chronic kidney disease develop protective antibodies against hepatitis B after vaccination, only 50-60% of patients with ESRD seroconvert ^9^.”

Reviewer #2 (Recommendations for the authors):To strengthen this manuscript, the reviewer has some suggestions below:1. For publication in eLife Journal, a bigger sample size of both cohorts and matching the number of candidates of different races would be appreciated, although the authors described this weakness in the MS.

We agree with the reviewer that this study would be strengthened with a larger number of subjects and better matching of race. We have added the limitation of the differing racial profiles between the control and HD cohorts in our discussion. Unfortunately, the addition of subjects and samples would be challenging at this time. We have included the following text:

“Our study is limited by different early time points between controls and HD (Day 1 vs Day 2 after each vaccination dose), by sample size particularly when subdividing SARS-CoV-2 history, and by differing racial profiles of the control and HD cohort.”

2. Setting up a validation cohort, either internal or external. Add data or discuss this limitation.

We have added this limitation to our discussion:

“Our small sample size further precludes a validation cohort to validate the study’s results. Future studies are needed for more comprehensive characterization of the immune pathway recruitment in response to the Covid-19 vaccinations in this population.”

3. Stratifying DM/hypertension/Dyslipidemia patients to compare with those without metabolic diseases in the HD cohort and see how metabolic diseases cause different immune responses in HD patients.

We agree that it would be illuminating to characterize immune differences stratified by etiology of end stage renal disease, as well as by time spent on hemodialysis. We attempted to stratify by DM/hypertension, and to identify differences as a function of months spent on hemodialysis. However, these did not yield meaningful/significant results, most likely d/t lacking sample size. We now include this in our discussion: “It will be of further interest to characterize immune responses dependent on etiology of ESRD (i.e. hypertension vs diabetes mellitus), as well as amount of time spent on hemodialysis, as this study was not sufficiently powered to disentangle these differences.”

4. Do these immune response differences also related to the years and the styles of HD?

All subjects in the HD cohort received maintenance hemodialysis three times a week, none used peritoneal dialysis. Please see response to #3 above.

5. The DM and hypertension candidates in the control cohorts could be deleted. Or the authors need to discuss the limitations in terms of comparison.

We purposefully did not exclude these conditions from our control cohort, as we did not wish for results to be driven by the immune dysregulation of these conditions. However, we do recognize the trade-off with this decision. We would in fact expect more significant findings using a control cohort with these exclusions. We have included this additional text in the discussion:

“We also cannot exclude the likelihood that these results are driven in part by immune dysregulation from diabetes mellitus, hypertension, and/or hyperlipidemia. While we did not exclude these in recruitment of our control cohort, subjects in the HD cohort are inevitably disproportionately affected by these conditions.”